# The Role of YAP and TAZ in Angiogenesis and Vascular Mimicry

**DOI:** 10.3390/cells8050407

**Published:** 2019-05-01

**Authors:** Taha Azad, Mina Ghahremani, Xiaolong Yang

**Affiliations:** 1Department of Pathology and Molecular Medicine, Queen’s University, Kingston, ON K7L 3N6, Canada; taha.biology@gmail.com; 2Department of Biology, Queen’s University, Kingston, ON K7L 3N6, Canada; m.gh.nbt@gmail.com

**Keywords:** Hippo pathway, cancer, angiogenesis, vascular mimicry, YAP, TAZ, MST1/2, LATS1/2

## Abstract

Angiogenesis, the formation of new blood vessels from pre-existing vasculature, is a physiological process that begins in utero and continues throughout life in both good health and disease. Understanding the underlying mechanism in angiogenesis could uncover a new therapeutic approach in pathological angiogenesis. Since its discovery, the Hippo signaling pathway has emerged as a key player in controlling organ size and tissue homeostasis. Recently, new studies have discovered that Hippo and two of its main effectors, Yes-associated protein (YAP) and its paralog transcription activator with PDZ binding motif (TAZ), play critical roles during angiogenesis. In this review, we summarize the mechanisms by which YAP/TAZ regulate endothelial cell shape, behavior, and function in angiogenesis. We further discuss how YAP/TAZ function as part of developmental and pathological angiogenesis. Finally, we review the role of YAP/TAZ in tumor vascular mimicry and propose directions for future work.

## 1. Introduction

### 1.1. Angiogenesis and Its Roles in Development and Pathogenesis

#### 1.1.1. Angiogenesis

Angiogenesis is the formation of new blood vessels from pre-existing vasculature. There are two main types of angiogenesis: sprouting angiogenesis and intussusceptive angiogenesis [1,2]. In sprouting angiogenesis, enzymatic degradation of the capillary basement membrane weakens the endothelial cell–cell interaction, which allows endothelial cells to proliferate and sprout toward an angiogenic stimulator. During this stage, endothelial cells flatten to make a long, slender, tapering pseudopodium, called filopodia; these are structures with a highly expanded endoplasmic reticulum and Golgi apparatus, high in proteolytic enzyme production. Proteolytic enzymes secreted from filopodia cleave the extracellular matrix and open a pathway for the developing sprout [3]. Intussusceptive angiogenesis, on the other hand, does not rely on endothelial cell proliferation or migration. In this type of angiogenesis, which is considered a fast angiogenesis compared to sprouting angiogenesis, an existing vessel is split into two new vessels just by cellular reorganization. Based on recent studies using scanning electron micrographs of Mercox casts, it has been shown that both types of angiogenesis occur frequently [4]. However, as the intussusceptive angiogenesis process is faster, it normally occurs in the early stage of angiogenesis compared to sprouting angiogenesis.

A variety of growth signaling pathways regulate angiogenesis (Figure 1). So far, vascular endothelial growth-factor A (VEGF-A) is the best-understood pro-angiogenic signal. Loss of the single allele or minor inhibition of this ligand is lethal during the developmental angiogenesis. VEGF-A binds to two receptor tyrosine kinases (RTKs): vascular endothelial growth factor receptor 1 (VEGFR1/Flt1) and VEGFR2 (KDR). Usually, VEGF-A induces signaling pathways related to angiogenesis. For example, it increases the secretion of matrix-metalloproteinases (MMPs) as well as endothelial cell proliferation [5]. Interestingly, it seems that upstream signaling pathways of angiogenesis are highly redundant and several other RTKs such as the fibroblast growth factor receptor (FGFR), ephrin, and platelet-derived growth factor receptor (PDGFR) can show the same effects as VEGFRs. Two very similar outputs of these signaling pathways are: (1) hypoxia-inducible factor 1 (HIF-1) transcriptional activator hetrodimerization, which usually occurs under low oxygen conditions and is involved in the adaptation to both cellular and organismal hypoxia; and (2) MMP activation, which causes degradation of the extracellular matrix to release more growth factors and to make it easier for endothelial cells to migrate.

#### 1.1.2. Angiogenesis in Development

Angiogenesis plays critical roles in embryogenesis and cardiovascular maturation [6,7]. The cardiovascular system is the first organ formed during vertebrate development. This complex vascular network provides different tissues with nutrition and immune system support. In primitive animals such as the fruit fly Drosophila melanogaster and the worm Caenorhabditis elegans, oxygen diffuses through their skin to all cells in their body. During evolution, animals became more complex and larger in size. Generally, modern metazoans have low surface area to volume ratio, which makes it impossible to distribute necessary oxygen and other nutrients using simple diffusion [8]. As oxygen diffusion is limited to about 100–200 μm thickness, all multicellular animals beyond this size need to recruit blood vessels by angiogenesis so that oxygen and nutrients can be distributed to their cells. Many cells with different properties and origins as well as numerous small angiogenic molecules are involved in this highly regulated process.

Shortly after gastrulation, angioblasts (endothelial progenitor cells) start differentiating from mesoderm layer in embryo. Angioblasts then unite together and form a very primitive network. This process is the first stage of the formation of the vascular network and is called vasculogenesis [9]. After uniting, angioblasts produce different growth factors to recruit other none-endothelial cells such as pericytes and smooth muscle cells to the primitive vasculature. The main role of non-endothelial cells is deposition of a specific basement membrane, for stabilization of the new made vasculature [9]. This primitive blood vessel network can be a foundation for more new vessel formation, at this point called angiogenesis. In summary, vasculogenesis is de novo vascular network formation in the embryo, whereas angiogenesis results in new blood vessels from pre-existing vessels.

VEGF-A and its receptor tyrosine kinase is the best-known angiogenic factors involved in angiogenesis. VEGF belongs to a gene family that includes several isoforms, namely VEGF-A, B, and C [10]. We will talk more about VEGF isoforms and their specific function in the following sections. Briefly, the VEGF-A gradient activates tip cells at the site of angiogenesis. These cells are very motile and guide the angiogenesis towards the angiogenic factors [11]. The Notch signaling pathway plays important role in regulation of VEGF-A response in endothelial cells to prevent excess angiogenesis. Adjacent stalk endothelial cells have high expression of Notch 1 and Notch 4. VEGF-A up-regulates the expression of delta-like ligand 4 (DLL-4) in tip cells, which binds to its receptor Notch 1 and Notch 4, and down-regulates VEGFR-2 expression in these cells [12]. At a short distance from tip cells, a critical step in the development of blood vessels and lumen formation and extension occurs. As it is difficult to mark the initial steps in tube formation, there is a lot of debate about its mechanism [13].

At the same time of angiogenesis the vessel start specifying into arteries, capillaries, and veins. It is assumed that the difference in blood flow is the main factor responsible for establishing arterial versus venous identity. It has been shown that the pattern of Eph receptor expression is different in arteries, capillaries, and veins. However, it is not clear whether or not the difference in Eph receptor expression is the result of arterial and venous formation or the cause [14]. Another important process happens is vascular anastomosis, which generates connection between angiogenic sprouts and blood vessels. Generally, there are two types of anastomosis, head-to-head and head-to-side. In head-to-head anastomosis, two tip cells and in head-to-side one tip cell and one functional blood vessel generate connections [13].

#### 1.1.3. Angiogenesis in Pathogenesis

Angiogenesis needs to be accurately controlled during development and adulthood. Therefore, not surprisingly, an imbalance in angiogenesis contributes to many diseases, including cancer, age-related macular degeneration (AMD), diabetic retinopathy, immune disorders, inflammation, infection, and ischemia [15,16,17]. So far, the majority of the reports on angiogenesis in human diseases come from studies on the roles of angiogenesis in tumorigenesis. There is much evidence indicating a dependency of tumor growth and metastasis on neovascularization occurring within and surrounding a tumor; normally, tumors reach a “steady state” size of roughly 2 mm diameter when not fed by neovascularization [18]. In addition to supporting tumor growth, blood vessels offer a means for cancer cells to metastasize away from the primary tumor. Tumor cells survive in the circulation, target new organs, and induce angiogenesis where they seed, perpetuating a vicious cycle. After embryogenesis, endothelial cells in vasculature become largely quiescent and only activated in certain circumstances, such as wound healing. During tumor expansion, however, the angiogenic switch is turned on, activating quiescent endothelial cells to sprout new vessels [19]. Mounting evidence suggests that the angiogenic switch is activated when tumors cause an imbalance in ratio of pro- and anti-angiogenic molecules. Usually, tumors increase the pro-angiogenic factor in different ways. For example, increasing tumor size reduces the tumor oxygen exchange rate, which activates the hypoxia signaling pathway. As result of this activation, the expression of HIF increases, which subsequently results in upregulation of pro-angiogenic molecules VEGF and FGF, and MMPs to increase angiogenesis (Figure 1). In addition, it has been shown that up-regulation of some oncogene signaling pathways such as PI3K and MAPK can contribute to increasing or reducing angiogenesis regulators. In addition, increasing MMPs and other matrix degrading enzymes in aggressive tumors can also contribute to angiogenesis by releasing such pro-angiogenic molecules from extracellular matrix as FGF and VEGF ligands, which are tethered to the extracellular matrix through their heparin-binding motif. Moreover, thrombospondin 1 (TSP-1), which binds and suppresses transmembrane receptors in endothelial cells, is an important counterbalance in the angiogenic switch. However, the signaling pathways and mechanisms in tumors that down-regulate TSP-1 are not well known [20].

Blood vessels within tumors have distinct features different from those of normal blood vessels. Their blood flow is not as consistent as that in normal blood vessels, which causes micro-hemorrhage and sometimes bleeding in the tumors. Furthermore, these vessels are very active in branching and sprouting new vessels. Metabolism, cell proliferation, and even cell death in the NE vasculature endothelial cells are very high in tumor vessels [21]. Since the endothelial cells are almost always the genetically intact cells from somatic quiescent cells, such different vessel morphology and endothelial cell behavior and metabolism are likely the result of tumor cells that create an imbalance in angiogenic molecule levels [21]. Even though some signaling pathways related to VEGF, FGFR, and HIF-1 have been studied very well, many questions remain to be answered.

### 1.2. Yes-Associated Protein (YAP) and Paralog Transcription Activator with PDZ Binding Motif (TAZ)

YAP and its paralog are WW domain-containing transcriptional co-activators and major effectors of the Hippo signaling pathway, which regulates tissue homeostasis, organ size, cell death, stem cell renewal, and mechanotransduction [22,23,24,25,26]. The final output of the Hippo pathway is the inhibition of YAP/TAZ co-transactivational activity by large tumor suppressor (LATS)1/2 kinases, and thereby their cytoplasmic sequestration and ubiquitin ligase-dependent degradation (Figure 2) [27,28,29]. The four main components of the Hippo pathway, including Warts [30], Salvador [31], Hippo [32], and the adaptor protein Mats [33] were found during the screening of tumor suppressor genes in *Drosophila* [34]. Hippo, Sav, Wts, and Mats in *Drosophila* are conserved proteins and homologous to mammalian mammalian STE20-like protein kinas 1/2 (MST1/2), WW45, LATS1/2, and Mps one binder 1 (MOB1), respectively [35,36]. In mammalians MST1/2 serine/threonine (S/T) kinases play key role in the Hippo pathway, as it is able to phosphorylate and activate three other components, including LATS, MOB, and Salvador [37,38,39]. When LATS1/2 S/T kinases are activated, they bind to and phosphorylate YAP/TAZ at five different conserved HxH/R/KxxS/T (H, histidine; R, arginine; K, lysine; x, any amino acid) motifs, including YAP S127 and TAZ S89 [33,36,40,41]. LATS-dependent phosphorylation of YAP/TAZ produces an interaction site for phospho-protein-binding protein 14-3-3, which inhibits YAP/TAZ nuclear localization and its co-transactivation of downstream genes with transcription factors such as TEA domain family protein (TEAD) and AP1 (Figure 2).

YAP/TAZ play a critical role in regulating many cellular behaviors in response to various internal and external stimuli [42]. For example, YAP/TAZ have been identified as conserved mechanotransducers for sensing diverse mechanical cues such as shear stress, cell shape, and extracellular matrix rigidity, and translating them into cell-specific transcriptional programs [43]. Cell extra-cellular matrix conformational change and mechanical stresses activate Rho GTPase mediated actin polymerization. Filamentous actin (F-actin) inhibits LATS activity and induces YAP/TAZ nuclear localization (Figure 2). Junction proteins can also regulate YAP/TAZ localization and activity [25]. Merlin (protein of the neurofibromatosis 2 (*NF2*) gene) directly interacts with angiomotin (AMOT) and α-catenin to recruit LATS kinase to adherent junction. Cross phosphorylation between AMOT and LATS at adherence junction results in YAP/TAZ phosphorylation and cytoplasmic retention. Scribble is a scaffold protein which recruits MST and LATS to basolateral junction and cause the same outcome. Junctions protein can also regulate YAP/TAZ activity just by sequestering them. It has been reported that AMOT and α-catenin can physically sequester YAP/TAZ in tight and adherent junctions [44,45].

YAP/TAZ also respond to extracellular cues such as hormones and growth factors. It has been shown that serum-borne lysophosphatidic acid (LPA) and sphingosine 1-phosphophate (S1P) act through a group of G-protein coupled receptors (GPCRs), G12/13-coupled receptors, to induce cell proliferation and migration. YAP/TAZ are necessary for G12/13-coupled receptors induced function. Rho GTPase is the main connector of GPCRs and YAP/TAZ. In addition, it has been discovered that epinephrine and glucagon can also regulate YAP/TAZ through a similar pathway [46]. In addition to GPCRs, RTKs are other important cell membrane proteins that regulate YAP/TAZ function. Ligand binding induces RTK dimerization at the cell membrane [47]. Two kinase domains cross-phosphorylate each other, which causes increasing kinase activity. The activated kinase domains phosphorylate other sites and produce docking sites for intracellular signaling proteins. The activated RTK and signaling proteins form a signaling complex that broadcasts signals along other signaling pathways. It has been shown that PI3-kinase (PI3K), one of the main downstream signaling pathways of RTKs, induces YAP/TAZ nuclear localization through inhibition of LATS activity (Figure 2) [48,49]. Recently, we provided the first evidence that the Hippo pathway effectors TAZ and YAP are critical mediators of PI3K-induced mammary tumorigenesis and synergistically function together with PI3K in transformation of mammary cells [50].

## 2. Roles of YAP/TAZ in the Regulation of Endothelial Function during Angiogenesis

Angiogenesis is a complex process with a series of sequential events. Endothelial cells as building block of vasculatures play a critical role in this event. During early stage of angiogenesis, endothelial cells loosen their junctions with other cells, change their shape and increase their motility. Therefore, to gain a better understanding of angiogenesis, the regulation of endothelial cell shape and behaviors should be firstly studied. Endothelial cells can be in quiescent, proliferating, or differentiating state according to the stimuli they received from their environment. If endothelial cells are seeded into collagen-coated plates, they enter to a high proliferating state. However, soon after plating the same cells in Engelbreth–Holm–Swarm mouse sarcoma (matrigel), the cells stop proliferating and differentiate to tube-like structure within 8–12 h. The comparison of gene expression by endothelial cells in these two distinct states reveal several important regulators, such as CTGF, CYR61, and angiopoietin-2, all of which are known as transcriptional target of YAP/TAZ [51]. Significantly, many genes and signaling pathways have also been shown to regulate endothelial cell shape and behavior though YAP/TAZ.

Most in vitro cancer angiogenesis assays use endothelial cells. Bovine aortic endothelial cells (BAECs) and human umbilical vein endothelial cells (HUVECs) are two main cell types that are used in these assays. Although it is easy to harvest the aforementioned cells form blood vessels and then expand them in primary cell cultures, they do not represent high tumour blood vessel heterogeneity [52,53,54,55]. Furthermore, in vitro conditions rarely reflect the in vivo environment, as they cannot expose the cells to hemodynamic forces, which are necessary to activate several signaling pathways in endothelial cells. Due to these limitations, many scientists are focusing on developing new methods to study the function of key component of angiogenesis. Recently, Cuia et al. [56] developed a three-dimensional (3D) microfluidic angiogenesis model to study immune-vascular and cell-matrix interactions. By using this model they demonstrated that soluble immunosuppressive cytokines and endothelial–macrophage interactions are involved in regulating glioblastoma tumor angiogenesis. It was observed that endothelial–macrophage interactions regulate in vitro proangiogenic activity through integrin receptors and Src–PI3K–YAP signaling. In another study, a 3D culture model was developed in auxetic scaffolds to evaluate vascular differentiation [57]. By using this model it was confirmed that cytoplasmic YAP expression of cells from auxetic scaffolds contributes to the enhanced vascular differentiation and angiogenesis. Therefore, developing new models, which are closer to in vivo models, will facilitate uncovering unknown functions of the YAP/TAZ and the Hippo pathway in angiogenesis.

### 2.1. Angiomotin (AMOT) Family

The AMOT family contains three members: AMOT, AMOT like 1 (AMOTL1), and AMOT-like protein 2 (AMOTL2). AMOT is expressed in two different isoforms through alternative splicing, p130-AMOT, and p80-AMOT (lacks the N-terminal 400 amino acids present in the p130 isoform). It is a membrane associated protein that regulates endothelial cells motility and behavior during angiogenesis [58]. p130-AMOT contains PPXY motifs, which mediate its interaction with the WW domain of YAP/TAZ [44]. p130-AMOT can inhibit transactivating function of YAP/TAZ by either bringing MST, LATS, and YAP to tight junctions and increasing YAP phosphorylation or physically sequestering YAP into F-actin. AMOT like 1 (AMOTL1) has 62% homology with AMOT and compensates for the absence of AMOT in mice models and exhibits similar expression pattern, controls endothelial polarity and junction stability during sprouting angiogenesis [59]. Studying AMOTL1 uncovered a novel function for the first time for cytoplasmic YAP. The E3 ubiquitin ligase Nedd4 targets AMOTL1 for ubiquitin-dependent degradation at tight junctions. Interestingly, YAP recruits c-Abl to tight junctions to phosphorylate E3 ubiquitin ligase Nedd4 on tyrosine. This tyrosine phosphorylation reduces the affinity of Nedd4 to AMOTL1, resulting in increasing the AMOTL1 and tight junction stability [60]. Surprisingly, another component of the Hippo, LATS can also regulate AMOT. LATS regulates AMOT function through direct phosphorylation on the LATS substrate conserved sequence, H/X/R/XXS. This phosphorylation site is S175 of AMOT, and has critical role in interaction with F-actin. When LATS phosphorylates Ser-175, it disrupts AMOT interaction with F-actin and reduces F-actin stress fibers and focal adhesions, which subsequently inhibits endothelial cell migration in vitro and zebrafish embryonic angiogenesis in vivo [61]. Since LATS is the major regulator of YAP/TAZ and AMOT and its interaction with Hippo signaling pathway can also regulate cell shape and motility of non-endothelial cells, which play important roles in cancer biology [62], how the LATS–AMOT interaction can affect YAP/TAZ function in angiogenesis is a critical question for future studies (Figure 3).

### 2.2. CD44

Vascular barrier integrity is another factor, which is mainly developed and regulated by endothelial cells during angiogenesis. This semi-selective barrier is disrupted in several diseases such as inflammation, atherosclerosis, and tumor angiogenesis. Endothelial cell-surface CD44 serves as a barrier regulatory receptor in endothelial cells [63]. Also, hyaluronan, a frequent glycosaminoglycan in the extracellular matrix, plays critical roles in angiogenesis, mainly through CD44 [64]. CD44 is also known for the regulation of matrix metalloproteinase level and activation, interactions with cortical membrane proteins, both of which plays a key role during angiogenesis. CD44 knock out mice show impaired barrier function, altered junctional organization, and dysfunctional endothelial cell junctions. It was observed that in these mice after angiogenesis, blood vessels express reduced levels of CD31 (also known as PECAM-1). Expression of CD31 restores barrier strength and endothelial cell morphology, which together with other evidence showing that CD44 regulates vascular endothelial barrier integrity and morphology via a CD31-dependent mechanism [65]. CD44 has been associated with one of upstream regulators of the Hippo pathway, Merlin [66]. Furthermore, CD44 deficiency causes reduction in CD31 and vascular endothelial (VE)-cadherin that has been connected with a reduction in tight junction, leading to Ajuba inhibition of LATS [67]. Another study showed independently that reduced CD31 expression induces Ajuba expression, resulting in LATS inactivation and YAP nuclear localization in cells originally derived from brain microvascular endothelial cells [68]. Additionally, it has been shown that CD44 regulates endothelial cell proliferation and survival by modulating cell adhesion molecules via the Hippo pathway [69,70]. Although these studies have shown CD44 as an upstream regulator of angiogenesis and the Hippo signaling pathway, how the CD44–Hippo interaction can affect YAP/TAZ function in angiogenesis is a critical question for future studies (Figure 3).

### 2.3. Extracellular Matrix (ECM)

One of the important factors that regulates endothelial cells’ behavior in normal condition and during angiogenesis is the physical property of the microenvironment. This includes cell morphology, ECM stiffness or confined adhesiveness, and flow dependent endothelial cells regulation. In normal conditions, increasing cell density causes YAP/TAZ cytoplasmic localization. When the cells are confluent their morphology is different and covers less area. To answer the question whether YAP/TAZ localization is regulated by cell morphology, micropattern plates have been used. Each of these plates has many microdomains with different sizes and shapes. Culturing cells in these plates showed that in small domains, where cells were round, YAP was mostly cytoplasmic, whereas on larger domains, where cells were spread and flat, YAP was in nucleolus [71]. Endothelial cell spreading and ECM stiffness induce YAP/TAZ nuclear localization independent of the NF2/Hippo/LATS pathway, instead requiring Rho activity and the actomyosin cytoskeleton [72]. Nakajima et al. [73] studied the spatiotemporal localization and transcriptional activity of YAP in endothelial cells of in vivo and found blood flow regulates localization of YAP through mechanotransduction signaling during physiological condition and angiogenesis. They found LATS-independent YAP nuclear localization in HUVEC cells exposed to shear stress and endothelial cells in blood vessels with flow. They also found, during vessel remodeling (a type of intussusceptive angiogenesis), that YAP localized into the nucleus through a flow-regulated mechanism. Recently, by using micropatterns a new technique has been designed to mimic mechanosensitivity and study the influence of extracellular physical cues on endothelial mechanosensing during angiogenesis. Using, this novel method, they found potential divergent kinetics for two important transcriptional related proteins in endothelial cells, MRTF-A and YAP, during angiogenesis (Figure 3) [74].

### 2.4. Transforming Growth Factor-β (TGFβ)/Bone Morphogenic Protein (BMP)

Several signaling pathways play critical role in endothelial cells during angiogenesis. The most well-known one is VEGF/VEGFR pathway. We will discuss in detail about this pathway and its links to the Hippo pathway in the following sections. Transforming growth factor-β (TGFβ)/bone morphogenic protein (BMP) regulates proliferation, differentiation, migration, and survival depending on the cell type. In endothelial cells, TGFβ1, TGFβ2, and TGFβ3 are three most common ligands for serine/threonine kinase receptors, including TGFβ type II receptor, activin receptor-like kinase (ALK1) and ALK5. The latter two are known as TGFβ type I receptors. TGFβ receptor activation causes specific SMAD transcription factors phosphorylation and nuclear localization. In the nucleus SMAD interacts with other transcription factors, which leads to the transcription of a wide array of target genes. Several in vivo studies have shown the importance of the TGFβ signaling pathway in angiogenesis [75,76,77]. Interestingly, the Hippo pathway interacts with TGFβ pathway in several points. The RASS1FA scaffold protein regulates the TGFβ-induced YAP–SMAD interaction and SMAD cytoplasmic retention. However, when RASSF1A is degraded, TGF-β activation causes YAP/SMAD nuclear localization and gene transcription [78]. Also, in response to hypoxia and TGF-β stimuli, LATS is ubiquitinated and degraded, resulting in YAP dephosphorylation and subsequently activation [79]. Endoglin is a type III TGFβ receptor that is upregulated in endothelial cells during angiogenesis. It is shown that Endoglin crosstalk Hippo pathway that leads to the regulation of secreted matricellular proteins and chemokines, cell migration and morphology, and extracellular matrix remodeling [80]. Also, another transcriptomics study showed the importance of the Hippo signaling pathway in TGFβ-activated cells. Evaluation of long non-coding RNAs (lncRNAs) and mRNAs with the Arraystar Human lncRNA Expression Microarray after TGFβ in human endothelial cells showed many of the transcriptome changes can be through YAP/TAZ [81]. However, how the interaction between TGFβ and YAP/TAZ in human endothelial cells can regulate angiogenesis is unclear (Figure 3).

### 2.5. WNT Pathway

The WNT signaling pathway has been widely implicated in maintenance of the vascular system and angiogenesis. Wnt binds to the frizzled (Fz) and low-density lipoprotein receptor-related protein (LRP) complex at the cell surface. Cytoplasmic β-catenin levels are regulated by complex protein down-stream of Fz/LRP that include dishevelled (Dsh), glycogen synthase kinase-3β (GSK-3), axin, and adenomatous polyposis coli (APC). When Wnt binds Fz/LRP, β-catenin degradation is inhibited by GSK-3/APC/axin. Inversely, in the absence of Wnt protein, β-catenin is targeted for ubiquitination. β-catenin can regulate many features of endothelial cells such as gene transcription and cytoskeleton structure [82]. Fz can also interact with the heterotrimeric G proteins and led to induction of YAP/TAZ transcriptional activity during central nervous system angiogenesis (Figure 3) [83]. Activation of heterotrimeric G proteins, specifically Gα12/13, can activate Rho GTPases resulting in inhibition of LATS1/2 to promote YAP/TAZ activation and TEAD-mediated transcription [84]. The Wnt antagonist Dickkopf2, known as regulated by YAP, promotes angiogenesis in rodent and human endothelial cells through regulation of filopodial dynamics and angiogenic sprouting via LRP6-mediated APC/Asef2/Cdc42 signaling [85]. This evidence shows the potential collaboration between Wnt and Hippo pathway to regulate angiogenesis (Figure 3).

### 2.6. PROX1 Transcriptional Programing—Key Regulators of Blood Vessel and Lymphatic Endothelial Cell Trans-Differentiation

The lymphatic microvasculature collects the removal of interstitial fluid/proteins and brings them back to the circulatory system. Although its function is in close coordination with blood vasculature, lymphatic endothelial cells have their own specific function and molecular composition. Isolation of primary lymphatic and blood microvasculature endothelial cells from human has revealed some similarities as well as unique molecular properties that distinguish lymphatic and blood vascular endothelium from each other [86]. Several studies showed that both lymphatic and blood vessel endothelial cells have the ability to monitor and respond to shear stress through YAP/TAZ to control cell–cell junction, and prevent regression of valves and focal vascular lumen collapse [73,87,88,89,90]. Both blood vessels and lymphatic endothelial cells have the same origin. Differentiation of lymphatic endothelial cells and having different fate from blood vessel endothelial cells mainly depends on a transcription factor activity, called Prox1 [91]. Prox1 is involved in cell fate determination, gene transcriptional regulation, and progenitor cell regulation during lymphogenesis as well as trans-differentiation of blood vessel and lymphatic endothelial cells. It has been shown that Prox1 expression is negatively regulated by YAP/TAZ activity [92]. YAP/TAZ regulate lymphatic identity through Prox1 transcriptional programing, which is a critical checkpoint underlying lymphatic endothelial cells differentiation from blood vessel endothelial cells [92].

## 3. Roles of YAP/TAZ in Developmental Vasculogenesis and Angiogenesis

Vasculogenesis and angiogenesis are responsible for forming new blood network in the embryo. Vasculogenesis begins with the differentiation of precursor cells (angioblasts) into endothelial cells and the de novo formation of a primitive vascular network, which makes the heart and the first primitive vascular plexus inside the embryo. Then, angiogenesis remodels and expands this network [93]. At early stage of mouse embryo development, TAZ has no expression, and comes up at all stages after blastocyst. However, YAP expression is dynamic and widespread during mouse development from the beginning. Knock out mice reveal the importance of YAP for early development of both yolk sac vasculature and placenta [94]. YAP-knocked out embryos die in the first half of gestation because of severe defects in early yolk sac vascular plexus and placenta development. Interestingly, TAZ cannot compensate this YAP specific phenotype. YAP function in vascular development has been also explored in zebrafish. YAP null mutant zebrafish showed increased vessel collapse and regression [74]. During regression, blood circulation in the endothelial cells activates YAP/TAZ, which leads to induction of CTGF and actin polymerization [95,96,97,98,99].

Comparing sprouting front, remodeling plexus, arteries, and veins in postnatal mouse retinas showed YAP/TAZ have distinct expression and localization in endothelial cells according to their physiological conditions and developmental stage [100,101,102,103,104,105,106,107]. YAP is expressed evenly throughout the vasculature with mostly cytoplasmic localization. During angiogenesis, YAP localizes into nucleus at sprouting area. TAZ expression is variable throughout the vasculature with highest expression in sprouting fronts. Surprisingly, when YAP or TAZ were conditionally knocked out in endothelial cells, mild vascular defects were observed [100]. Deleting both YAP and TAZ in mice causes a dramatic defect in blood vessel development at the retinal vasculature, a 21% decrease in radial expansion, a 26% decrease in capillary density, and a 55% decrease in branching frequency. These results indicate that endothelial YAP and TAZ are critical for the vascular growth, branching, and regularity of blood vessel network. In another study, it was observed that overexpression of YAP-active form increased angiogenic sprouting, which is blocked by angiopoietin-2 (ANG-2) depletion or soluble Tie-2 treatment [101].

YAP and TAZ are known mainly for their co-transcriptional activity in the nucleus. However, cytoplasmic YAP/TAZ have also distinct functions in angiogenesis and vascular remodeling. Deletion of the LATS1/2 in mice endothelial cells results reduction in YAP/TAZ phosphorylation and cytoplasmic retention. The retinas of these mice show migration defect with reduced extension distance, which can be partially rescued by overexpressing YAP-S127D (cytoplasmic form of YAP). Intriguingly, deletion of YAP/TAZ, Cdc42, or LATS1/2 in endothelial cells has a very similar phenotype in the retina, implying the possibility of operation in a common pathway in angiogenic tip cell development. Cytoplasmic YAP/TAZ positively regulates the activity of the Rho family GTPase CDC42 activity, which causes proliferation, and migration of vascular endothelial cells during retinal angiogenesis [102].

VEGFRs and their ligands VEGFs regulate the functions of several types of cells in cardiovascular system such as hematopoietic precursors, as well as endothelial and lymphendothelial cells. In mammals, there are three RTKs, known as VEGFR1–3, which bind to five VEGF ligands, derived from alternative splicing. VEGFR-1 is one of main negative or positive regulators of VEGFR-2. VEGFR-2 and VEGFR-3 play crucial roles in angiogenesis and lymphogenesis, respectively. VEGF family members secrete 40-kDa dimeric glycoproteins including VEGFA, B, C, D and placenta growth factor (PLGF) [103]. PLGF and VEGFB just bind VEGFR-1. VEGFC, and D are known ligands for VEGFR-3 and regulate lymphogenesis. VEGFA, and B bind to both VEGFR-1 and -2 and are involved in regulating of endothelial cell mitosis, and survival, as well as angiogenesis and microvascular permeability. Alternative gene splicing of VEGFA makes several isoforms namely, VEGF_121_, VEGF_165_, VEGF_189_, and VEGF_206_ [104]. Ligand binding induces VEGFR dimerization and increasing kinase domain activity. Cross phosphorylation between kinase domain produces docking site for intracellular signaling protein that broadcast signaling. PI3K and MAPK are two main signaling pathways downstream of VEGFRs. Recently, we made a novel bioluminescent-based biosensor to quantify the kinase activity of the large tumor suppressor (LATS), a central kinase in the Hippo signaling pathway [105,106]. Using this biosensor against a small kinase library, we found VEGFR activation by VEGF triggers PI3K/MAPK signaling, which subsequently activates the Hippo effectors YAP and TAZ (Figure 3). We demonstrated VEGF activation induces angiogenesis in vitro and in vivo, which was abolished by YAP/TAZ knockdown. In addition, we found treatment of YAP/TAZ knocked down cells with Cyr61 and ANG-2, known as YAP/TAZ pro-angiogenic targets, can partially rescue the phenotype. Wang et al. [107] identified that YAP/TAZ are essential co-transcriptional activators in endothelial cells and YAP/TAZ activity is controlled by VEGF during developmental angiogenesis. YAP/TAZ localization in endothelial cells is different during embryogenesis. In early embryos (E10.5 and E11.5) YAP/TAZ are predominantly in the nucleus. However, they are mostly localized in the cytoplasm in later developmental stages (E14.5). Interestingly, by using transgenic mouse cell line overexpressing VEGF, increased angiogenesis and nuclear YAP/TAZ were observed. Moreover, endothelia-specific deletion of YAP/TAZ caused severe vascular defects throughout the whole body as well as yolk sac vascularization. They also studied the roles of YAP/TAZ in retina and brain vascularization during postnatal angiogenesis. Deletion of YAP/TAZ in endothelial cells reduced cell proliferation but had no effect on blood vessel regression. However, YAP/TAZ knockout reduced vessel coverage, length, and branch points in the developing brain cortex. Significantly, they also showed that VEGF changes cytoskeleton dynamics by regulating Src family kinases (SFKs) and Rho GTPases, which results in YAP/TAZ nuclear localization. Interestingly, YAP/TAZ can also regulate VEGFR2 and its activity. YAP/TAZ knocked down in endothelial cell impairs proper VEGFR2 trafficking and its downstream signaling. ChIP-Seq analysis of endothelial cells after VEGF treatment and comparison with knocked down YAP/TAZ cells revealed that VEGF induces a YAP/TAZ-dependent transcriptome linked to cytoskeleton assembling genes.

Signal transducer and activator of transcription 3 (STAT3) is a transcription factor interacting with YAP/TAZ interacting proteins in endothelial cells [108]. Interleukin 6 (IL-6) treatment results in STAT3 phosphorylation by receptor-associated Janus kinase that induces its nuclear localization and regulates gene expression in the acute-phase inflammatory response. YAP binding extends IL-6-driven STAT3 accumulation in the nucleus and increases the expression of Ang-2, which promotes angiogenesis. Ang-2 blockage or selective STAT3 inhibitors attenuate retinal angiogenesis in Tie2Cre-mediated YAP transgenic mice.

Additionally, YAP/TAZ can also regulate endothelial cells and angiogenesis through regulating none-endothelial cells such as pericytes as well as vascular smooth muscle cells or mural cells. Pericytes are functionally important as if they are lost, blood vessels become hemorrhagic and hyper dilated, which cause edema, diabetic retinopathy, and embryonic lethality [109]. Recent studies have shown the importance of pericytes in tumor progression [110]. Kato et al. [111] showed recently that YAP/TAZ regulates several key functions of pericytes. Knocking down of YAP/TAZ in pericytes of mice reduces hepatocyte growth factor expression which leads to impaired activation of the c-Met receptor in alveolar cells [111]. They also showed YAP/TAZ regulates angiopoietin-1 expression in pericytes, which together with hepatocyte growth factor coordinates the behavior vascular cells during lung morphogenesis.

## 4. Roles of YAP/TAZ in Pathological Angiogenesis

In general angiogenesis related diseases can be categorized into two major groups depending on whether there is need for more angiogenesis or it should be inhibited to cure the disease. In the first category, therapeutic angiogenesis can help damaged tissue to be repaired, such as in limb ischemia, myocardial infarction, and arteriosclerosis. In second category, pathological angiogenesis should be inhibited to stop or slow down the progression of a disease such as in benign and malignant angiogenic tumors, as well as retinopathies [112].

One of the factors that can affect systematic angiogenesis is. Mesenchymal stromal cells (MSCs) are multipotent non-hematopoietic cells with multi-lineage potential to differentiate into various tissues, including osteoblasts (bone cells), chondrocytes (cartilage cells), myocytes (muscle cells), and adipocytes (fat cells which give rise to marrow adipose tissue). MSCs are also known to regulate angiogenesis in different organs especially in bone marrow. MSCs detect mechanosignals and regulate angiogenesis through secretion of several pro-angiogenic factors such as VEGF and placental growth factor (PGF). The amount of this pro-angiogenic factors secretion in MSCs cells is correlated with age. Interestingly, MSCs derived from children are more mechanosensitive, showing enhanced angiogenesis on stiff substrates. This difference is possibly due to activity in Hippo signaling as MSCs derived from children have lower Hippo signaling activity and higher YAP nuclear localization and transactivating activity [113]. Another factor affecting systematic angiogenesis is endothelial cell-derived exosomes (EDEs) that promote trans-endothelial migration and angiogenesis. Atherosclerotic cerebrovascular disease includes a variety of medical conditions such as elevated blood pressure [114]. Levels of YAP in EDE are higher and those of pS127-YAP are lower in patients with atherosclerotic cerebrovascular disease [114]. However, how this YAP protein level change in EDEs regulates angiogenesis is not completely understood.

Structural changes of the vascular wall and vascular remodeling is commonly correlated with hypertension. Vascular remodeling can be induced in mice by angiotensin II fusion in 2-weeks. Mice treated with angiotensin II show YAP up-regulation in their blood vessels. Interestingly, disrupting the YAP–TEAD interaction with verteporfin inhibits Ang II-induced vascular remodeling in mice [115].

Retinal neovascularization, such as age-related macular degeneration (AMD or ARMD), retinopathy of prematurity (ROP), and proliferative diabetic retinopathy (PDR), are the leading cause of blindness in developed countries. Although the mechanism of retinal neovascularization is not well understood, endothelial dysfunction as a result of hypoxic stimuli such as ischemia or inflammation is one of main players. Laser photocoagulation and VEGFR inhibitors are the two main existing treatments. Laser photocoagulation can destroy retinal neurons and affect normal visual function. Effects of VEGFR inhibitors are mostly short-term due to the development of resistance to anti-VEGF therapies [116]. Therefore, it is urgent to study the underlying cellular and molecular mechanisms of these diseases to find novel therapeutic approaches. AMD is the leading cause of sight loss, which can be classified as wet or dry. The wet AMD is characterized by choroidal neovascularization with abnormal blood vessels that grows behind the macula and leak fluid. The proliferation of choroidal endothelial cells is the key step in angiogenesis. The in vivo AMD mouse model is created by laser photocoagulation. In this model, YAP expression in choroidal endothelial cells is significantly higher and mostly localized in the nucleus. Choroidal endothelial cells with highly expressed YAP also express high protein level proliferating cell nuclear antigen (PCNA) with a higher mitosis rate. Intriguingly, knocking down YAP in these cells in combination therapy with ranibizumab, a VEGF monoclonal antibody, decreased the expression of PCNA, choroidal endothelial cell proliferation, and the incidence as well as leakage area of choroidal neovascularization [117]. YAP/TAZ conditionally knocked out mice show markedly attenuated choroidal neovascularization volume and suppressed vascular leakage compared with wild type mice [118]. Also, it has been shown that treatment of human retinal microvascular endothelial cells (HRMECs) with dimethyloxalylglycine (DMOG), to mimic hypoxia conditions, induces angiogenesis through YAP and STAT3 [119]. In endothelial cells and during hypoxia conditions total YAP, and p-STAT3 (Tyr705) increases, while pS127-YAP decreases. Un-phosphorylated YAP localizes p-STAT3 into the nucleus and promotes the transcription of VEGF. YAP also regulates proliferation, migration and tube formation of HRMECs in vitro. Additionally, YAP inhibition reduces retinal pathological neovascularization in mouse oxygen-induced retinopathy model.

High morbidity and mortality in diabetic patients are mainly due to impaired angiogenesis and wound healing. The most common complications of diabetes include having damaged blood vessels, increased permeability, and growth of new blood vessels in the retina. These symptoms together are known as diabetic retinopathy. Treatment for diabetic retinopathy is similar to AMD treatment and has low efficacy to arrest the progression of this disease. The Hippo signaling pathway is important for the progression of diabetic retinopathy and can be used to develop new therapeutic approaches. Diabetic rats show decreased levels of p-S127-YAP and p-MST, while TAZ and TEAD total proteins are increased in their retinas [120]. These changes in the Hippo signaling activity result in increasing pro-angiogenic factors secretion and retina new blood vessel formation. Another complication of diabetes is delayed wound healing due to impaired angiogenesis and poor dermal healing. Recently, it has been shown that substance P, a neurotransmitter composed of 11 amino acids, accelerates wound healing in diabetic mice through endothelial progenitor cell mobilization and YAP activation [121]. Additionally, a non-invasive method, low-intensity pulsed ultrasound can accelerate healing through activation of YAP and TAZ in endothelial cells. Low-intensity pulsed ultrasound treatment activates YAP and TAZ nuclear localization through LATS inhibition, which can promote angiogenesis and vascular remodeling [122].

Hyperglycemia and elevated free fatty acids in diabetic patients cause metabolic stress, which inhibits endothelial angiogenesis. The Hippo signaling pathway and YAP/TAZ play critical role in endothelial angiogenesis inhibition during metabolic stress. For example, treatment of endothelial cells with palmitic acid inhibits their cell proliferation, migration, and tube formation ability. After palmitic acid treatment, MST expression increases dramatically, which then activates LATS and results in YAP phosphorylation and exclusion from nucleolus. Interestingly, either knocking down MST or overexpression of YAP prevented palmitic-induced inhibition of angiogenesis. Palmitic acid treatment damages endothelial cells mitochondria and releases mitochondrial DNA. Mitochondrial DNA activates cytosolic DNA sensor cyclic GMP-AMP synthase (cGAS)-stimulator of interferon genes (STING)-interferon regulatory factor 3 (IRF-3). cGAS-STING-IRF3 signaling activates IRF3 transcription factor, which then directly interacts with MST promoter and increases its expression [123]. In addition to the mentioned functions, Hippo signaling also regulates endothelial metabolisms. Peroxisome proliferator-activated receptor gamma co-activator 1-alpha (PGC1α), a major player controlling glucose consumption in mitochondria, is regulated by YAP. Constitutively active YAP (YAP-S127A) up-regulates PGC1α expression, resulting in increased in vitro and vascular morphogenesis in the fibrin gel subcutaneously implanted on mice. Knocking down PGC1α inhibits YAP-S127A-induced angiogenesis [124].

## 5. Roles of YAP/TAZ in Tumor Angiogenesis and Vascular Mimicry

Like normal tissues, cancer cells need to have access to blood vessels to get oxygen and nutrition and evacuate their waste. Almost always, angiogenic switch activates quiescent vasculature near tumors to form new blood vessels. After angiogenic switch, tumors exhibit diverse patterns of neovascularization. Cancer cells in primary tumors use nearby vessels for entering to blood, known as intravasation. Those cells surviving in the blood extravasate and form micrometastases. Dormant micrometastases can form macroscopic tumors if they are able to activate tumor angiogenesis [125]. In the past decades many studies tried to find the mechanism by which cellular genes regulate angiogenesis during tumor progression. YAP and TAZ are known as two key regulators of tumor angiogenesis.

Astrocytoma, a type of cancer that can form in the brain or spinal cord, has very significant angiogenesis. Analysis of blood vessel density in diffuse astrocytomas, in particular glioblastoma, revealed that the levels of TAZ in endothelial cells are positively correlated with blood vessel density. TAZ expression is also correlated with vascular VEGFR2 level. Expression of both TAZ and VEGFR2 is up-regulated in endothelial cells of high grade glioblastoma, which indicates the importance of VEGFR2/TAZ signaling pathway in angiogenesis [126]. Comparing YAP and TAZ localization in normal brain and tumor-associated endothelial cells shows YAP and TAZ are highly accumulated in the nucleus [107]. Angiosarcoma is another type of cancers with high level of angiogenesis. Unexpectedly, YAP activation inhibits angiogenesis in angiosarcoma. Heterogonous angiosarcoma cells can be divided in to CD31 high or low expressing cells. CD31^low^ cells are more tumorigenic and chemoresistant than CD31^high^ cells due to more efficient reactive oxygen species (ROS) detoxification. CD31^low^ cells also have less angiogenesis ability with higher YAP nuclear localization [127].

YAP shows high activation in cholangiocarcinoma patient biopsies and human cholangiocarcinoma cell lines. Microarray expression profiling of cholangiocarcinoma cells with overexpressed or knocked-down YAP reveal that YAP regulates important genes in proliferation, apoptosis, and angiogenesis [128]. YAP/TEAD regulate pro-angiogenic microfibrillar-associated protein 5 (MFAP5) transcription in cholangiocarcinoma cells. Secreted MFAP5 induces angiogenesis of human micro-vascular endothelial cells. Also, YAP activation is correlated with high MFAP5 expression in both human cholangiocarcinoma and cholangiocarcinoma xenografts. Additionally, YAP activation and its correlation to angiogenesis have been reported in gastric cancer [129]. Cancer-associated MSCs are important component of tumor microenvironment. MSCs cells isolated from cancer patients demonstrate different features compared to bone marrow-derived MSCs. Cancer-associated MSCs have higher YAP expression. YAP knockout in MSCs inhibits their proliferation, migration, invasion, and pro-angiogenic ability by reducing the activation of β-catenin and its target genes.

Vasculogenic mimicry (VM) is a term that was first introduced by Maniotis et al. [130] in 1999 when they observed that tissue sections from aggressive human intraocular and metastatic melanomas do not show any significant necrosis and have patterned networks of interconnected loops of extra-cellular matrix. Vascular mimicry (VM) is the formation of blood supply system by tumor cells rather than endothelial cells, which is independent of typical modes of angiogenesis. High tumor VM reflects the plasticity of aggressive tumor cells to express vascular cell markers and tumor vasculature. High tumor VM is associated with a high tumor grade, shorter survival, invasion, metastasis and poor prognosis [131]. The first characterization of VM was in melanoma. In melanoma, VM is characterized as involving areas with high levels of laminin, collagen IV and VI, and heparin sulfate proteoglycans, and it should have plasma and red blood cells, confirming its function as a blood/nutrient supplier [132]. In addition to providing oxygen and nutrients, VM can provide a new route for escaping and metastasizing cancer cells. So far VM has been shown in variety of cancers including melanoma [133], breast and lung cancer [134,135], ovarian cancer [136], osteosarcoma [137], gastric cancer [138], bladder cancer [139], hepatocellular cancer [140], and colorectal cancer [141], etc.

Small population of cancer cells have stem cell characteristics, which are known as critical player in initiation, progression, metastasis, and recurrence of tumors. Cancer stem cells can be quiescent for several years to several decades. Upon activation, they can transdifferentiate to different tumor components including the vasculature. YAP is elevated in cancer stem cells derived from non-small cell lung cancer and regulates their ability to form angiogenic tubules [142]. YAP induces vasculogenic mimicry in cancer stem cells by regulating the embryonic stem cell transcription factor Sox2. YAP regulates Sox2 transcription independent of TEAD, through physical interaction with Oct4 by its WW domain. It has been shown that YAP also regulates vasculogenic mimicry of pancreatic ductal adenocarcinoma via interaction with TEAD [143]. YAP inhibition using verteporfin or short hairpin RNA (shRNA) suppresses vasculogenic mimicry by reducing Ang2, MMP2, VE-cadherin, and SMA expression in vitro and in vivo. Tube formation assay experiments showed when YAP is inhibited, pancreatic ductal adenocarcinoma cancer cells have less ability to induce HUVEC tube formation in vitro. Molecular analysis showed that in addition to Ang2 and MMP2, cyclin E1 decreased, which causes cell cycle arrest in the G1 phase. Also, poly-ADP ribose polymerase (PARP) western blot analysis showed more apoptotic cells in pancreatic cells. Evaluating of pancreatic adenocarcinoma xenograft mode with and without YAP inhibition showed reduction in tumour cells as well as VM content. In another study, we showed that YAP and TAZ regulate vasculogenic mimicry by upregulating pro-angiogenic factors such as Cyr61 and Ang2 in breast cancer cell lines [106]. In our study we inhibit YAP/TAZ by using verteporfin or short interference RNA (siRNA). Consistently with previous publication, YAP inhibition reduced VM in vitro. Also, we found that Cyr61 and Ang-2 may play critical roles in VM regulation, as an addback experiment using purified Ang-2 and Cyr61 can significantly rescue the phenotype.

## 6. Conclusion and Future Directions

In conclusion, YAP and TAZ play central roles in regulating angiogenesis during development and pathogenesis of many diseases (Table 1). YAP/TAZ can induce angiogenesis through activation of multiple downstream targets such as Cyr61, Ang2, and MMP2. YAP/TAZ may also mediate angiogenesis induced by CD44, ECM, TGFβ, AMOT, and RTKs such as VEGFR, FGFR, IGFR, TGFR, and Tie2 (Figure 3). On one hand, YAP/TAZ regulates key functions of endothelial cells and pro-angiogenic production of cancer cells during tumor angiogenesis. On another hand YAP/TAZ are involved in several important features of cancer cells such as cell proliferation, immune evasion, EMT, and the cancer stem cell phenotype [144]. Therefore, targeting YAP/TAZ offers a new and attractive strategy to treat aggressive tumors by targeting both angiogenesis and other cancer phenotypes at the same time. In addition, since targeting RTKs involved in angiogenesis (such as VEGFR, and FGFR) often results in short-term and limited effects on overall survival of cancer patients [145,146], combination treatments which target both RTKs and Hippo signaling can offer a new and attractive strategy to treat aggressive tumors. Moreover, since YAP/TAZ are also involved in the dysregulation of angiogenesis during the development of many other diseases such as diabetes and AMD, targeting YAP/TAZ will provide new therapeutic strategies to treat these diseases in the future.

## Figures and Tables

**Figure 1 cells-08-00407-f001:**
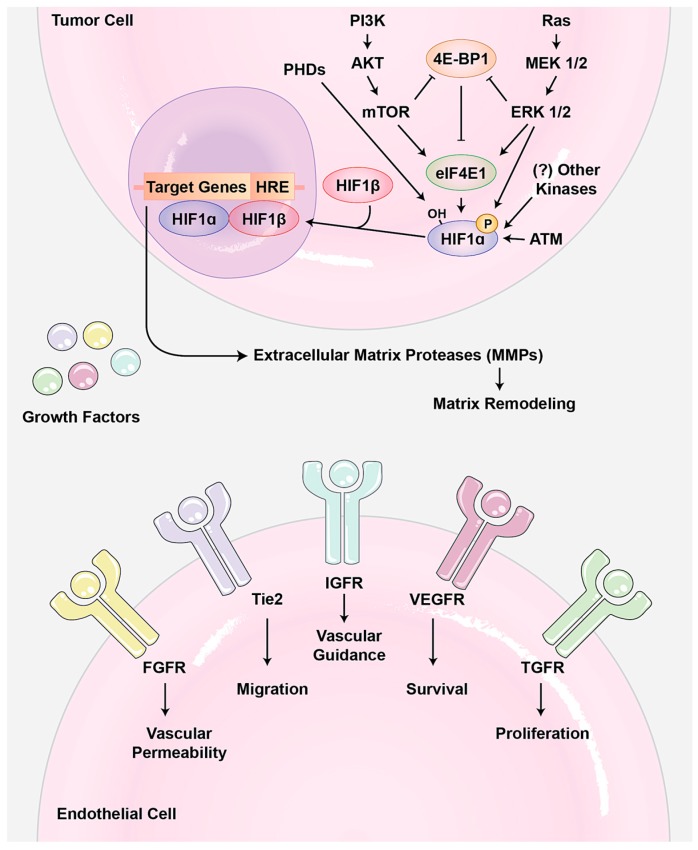
Tumor angiogenesis. In tumor cells, several signaling pathways such as phosphoinositide 3-kinase (PI3K) and mitogen-activated protein kinase (MAPK) can phosphorylate eIF4E1 to increase the HIF-1α translational level. Also, direct phosphorylation and hydroxylation are two important post-translational modifications that cause HIF-1 dimerization and promote its degradation, respectively. Activated HIF-1 then functions as a master switch to induce the expression of several growth factors such as VEGF, PDGF, and Ang2, as well as extracellular matrix proteases such as matrix metalloproteinase (MMP). MMPs allow the endothelial cells to escape into the interstitial matrix during sprouting angiogenesis. Also, secreted growth factors induce survival, proliferation, migration, and vascular permeability in endothelial cells. Abbreviations—PHD: prolyl hydroxylase domain protein; ATM: ataxia telangiectasia mutated; HRE: HIF-1 response elements; HIF-1: hypoxia-inducible factor 1; MMP: matrix metalloproteinase; VEGF: vascular endothelial growth-factor; VEGFR: VEGF receptor; PDGF: platelet-derived growth factor.

**Figure 2 cells-08-00407-f002:**
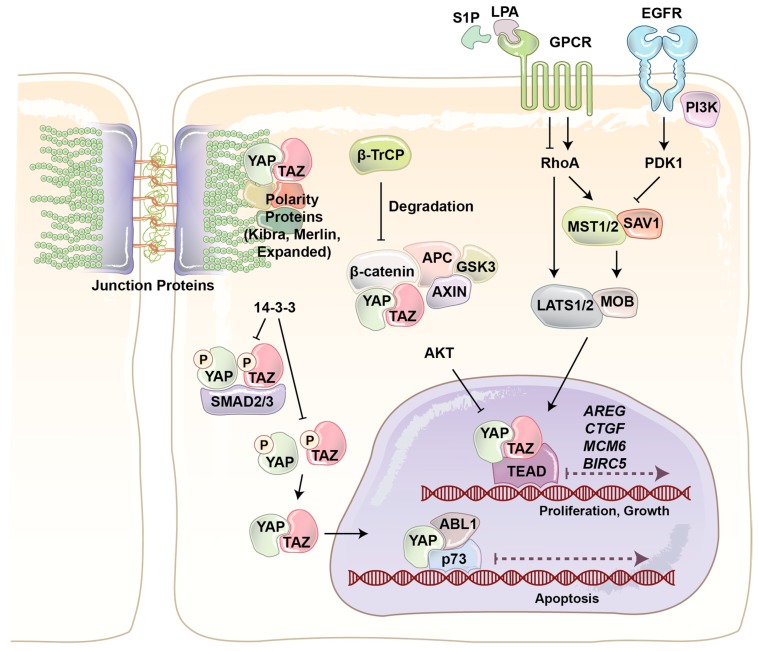
An overview of the regulation of YAP and TAZ transcriptional co-activators. YAP and TAZ are downstream mediators of numerous signaling pathways such as G-protein couple receptors (GPCRs) and epidermal growth factor (EGFR). YAP and TAZ localization is mainly regulated through phosphorylation by large tumor suppressor (LATS). The 14-3-3 phosphobinding protein interacts with and sequesters phosphorylated YAP and TAZ. YAP and TAZ localization is also regulated through physical interaction, for example with SMAD, β-catenin, and junction proteins. YAP: Yes-associated protein (YAP); TAZ: transcription activator with PDZ binding motif.

**Figure 3 cells-08-00407-f003:**
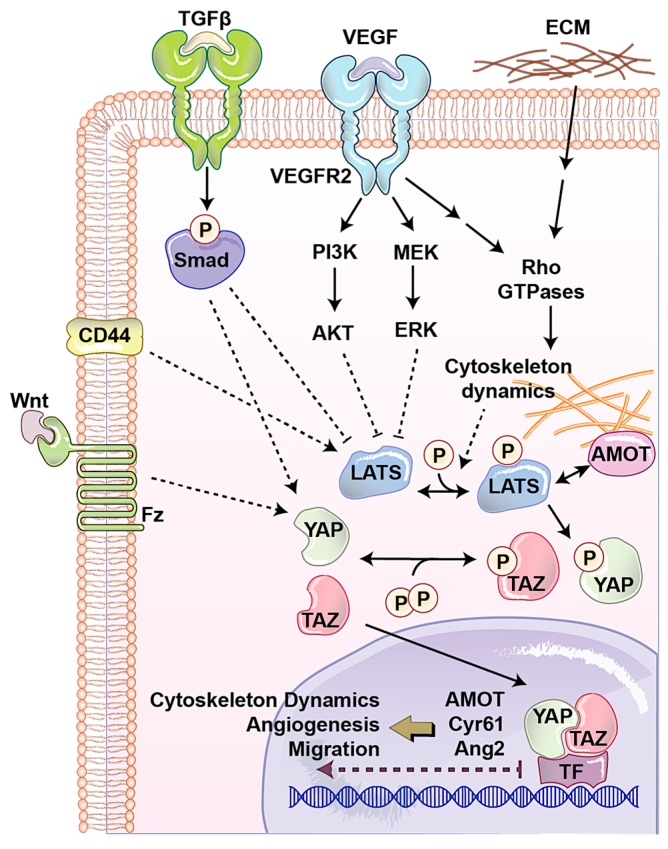
An overview of the regulation of YAP and TAZ transcriptional co-activators in angiogenesis. Several receptors regulate YAP and TAZ activity directly via LATS or other unknown pathways, which can affect angiogenesis. VEGFR regulates YAP and TAZ through three main pathways including the Rho GTPase, MAPK, and PI3K pathway. The TGF-β, Wnt, and CD44 pathways regulate YAP and TAZ as well as LATS activity through several not well-known mechanisms. Abbreviations—TF: transcription factor; ECM: extracellular matrix; Fz: frizzled receptor.

**Table 1 cells-08-00407-t001:** Summary of the role of YAP and TAZ function in physiological and pathological angiogenesis.

Main Finding	Ref.
Angiomotin is a negative regulator of YAP	[44]
Angiomotin-like 1 degradation by Nedd4 is regulated by YAP through c-ABL	[60]
YAP/TAZ are the main mediators of mechanotransduction in endothelial cells	[72]
BMP9 crosstalk with the Hippo pathway regulates endothelial cell matricellular response	[80]
mRNAs upregulated in ECs in response to TGFβ1 treatment are involved in hippo signaling	[81]
Flow-dependent endothelial YAP regulation contributes to vessel maintenance	[73]
YAP/TAZ negatively regulate prox1 during developmental and pathologic lymphangiogenesis	[92]
YAP disruption in mice causes defects in yolk sac vasculogenesis and chorioallantoic fusion	[94]
YAP/TAZ activity is essential for vascular regression via Ctgf and actin polymerization	[95]
Adherens junction and endothelial cell distribution in angiogenesis is regulated by YAP/TAZ	[100]
YAP regulates angiopoietin-2 expression in ECs	[101]
Vascular tip cell migration is regulated by YAP/TAZ-CDC42 signaling pathway	[102]
VEGFR is a regulator of YAP/TAZ in the Hippo pathway in angiogenesis through PI3K/MAPK pathways	[106]
VEGF activates YAP/TAZ via its effects on actin cytoskeleton	[107]
YAP promotes angiogenesis via Stat3	[108]
YAP mediates angiotensin II-induced vascular smooth muscle cell phenotypic modulation and hypertensive vascular remodelling	[115]
YAP inhibition ameliorates choroidal neovascularization	[117]
YAP/TAZ regulates vascular barrier maturation	[118]
YAP via interacting with STAT3 regulates VEGF-induced angiogenesis in retina	[119]
Substance P accelerates wound healing in type 2 diabetic mice through YAP activation	[121]
Ultrasound treatment accelerates angiogenesis by activating YAP/TAZ	[122]
Palmitic acid inhibits angiogenesis through YAP suppression	[123]
YAP1-TEAD1 controls angiogenesis and mitochondrial biogenesis through PGC1α.	[124]
Blood vascular density and VEGFR2 expression in astrocytomas is regulated by TAZ	[126]
Cell proliferation, chemoresistance, and angiogenesis in human cholangiocarcinoma is regulated by YAP	[128]
YAP regulates OCT4 activity and SOX2 expression to facilitate vascular mimicry	[142]
Verteporfin suppresses vasculogenic mimicry of pancreatic ductal adenocarcinoma	[143]

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
