# Peer review of "The Role of YAP and TAZ in Angiogenesis and Vascular Mimicry"

_cells, 2019, doi:10.3390/cells8050407_

Round 1

Reviewer 1 Report

In this review Azad et al reviewed the role of YAP and TAZ in angiogenesis.

Overall the review is well written and the signaling mechanism of YAP and TAZ is well demonstrated.

The figures summarize the signaling molecules interacting with YAP/TAZ. If authors include a table or figure to summarize the YAP/TAZ in physiological and pathological angiogenesis, that would help readers to understand the role of YAP/TAZ in angiogenesis. 

Author Response

Response to Reviewer#1

Comments and Suggestions for Authors

In this review Azad et al reviewed the role of YAP and TAZ in angiogenesis.

Overall the review is well written and the signaling mechanism of YAP and TAZ is well demonstrated.

The figures summarize the signaling molecules interacting with YAP/TAZ. If authors include a table or figure to summarize the YAP/TAZ in physiological and pathological angiogenesis, that would help readers to understand the role of YAP/TAZ in angiogenesis. 

We would like to thank the reviewer for the comments. We have included a table to summarize the roles of YAP/TAZ in physiological and pathological angiogenesis.

Reviewer 2 Report

General comment 

This review by Azad et al address the role of yap and taz in the vascular system during developmental as well as pathological conditions. 

While the review is in general interesting the title of the review doesn’t really reflect the content. The first part is indeed amply covered by the part of vascular mimicry is minor. As there are already some recent reviews converting the role of the hippo pathway and yap taz in angiogenesis (Elaimy et al. Sci Signal 2018; Zhou et al. J Immunol Res 2018; Park et al. BMB Rep 2018), it is recommended that in this review the authors focus deeper in vascular mimicry. 

Specific comments 

1. In general the text lacks references.  For example: 

* Line 110: …Since endothelial cells… this sentence should have references.

* Paragraph starting in line 337: the authors cite only refs 79 and 80 for describing the expression pattern of yap and taz in endothelial cells. However, other studies have also shown the expression of these cofactors in endothelial cells and thus should be fitted as well. 

* Line 432: first sentence lacks references. 

2. The role of yap taz in pericytes and in lymphatic endothelial cells should also be described. 

3. Line 126: a part of the sentence seems to be missing.

4. figure 2. polarity proteins and junction proteins should be defined and indicated with names. 

5. The section of angiogenesis in development is really small. As most of the function for yap and taz has been described during physiological developmental angiogenesis, this section should be expanded.  

6. the section of vascular mimicry should be expanded. 

Author Response

Response to Reviewer#2

Comments and Suggestions for Authors

General comment 

This review by Azad et al address the role of yap and taz in the vascular system during developmental as well as pathological conditions. 

While the review is in general interesting the title of the review doesn’t really reflect the content. The first part is indeed amply covered by the part of vascular mimicry is minor. As there are already some recent reviews converting the role of the hippo pathway and yap taz in angiogenesis (Elaimy et al. Sci Signal 2018; Zhou et al. J Immunol Res 2018; Park et al. BMB Rep 2018), it is recommended that in this review the authors focus deeper in vascular mimicry. 

Thanks the reviewer for the comments. Although vascular mimicry (VM) plays important role during tumour progression, it is a newborn area in the field with a limited knowledge, especially on the roles of YAP/TAZ in VM. However, in response to the reviewer’s comments, we have extended the section of VM and tried our best to cover all published papers on the roles of YAP/TAZ in TM and point to future directions.
      Although there are 3 short reviews published prior to the submission of our manuscript [1. Elaimy et al. 2018(VEGF-YAP/TAZ interaction in angiogenesis and cancer); 2. Zhou et al 2018 (Hippo pathway in cardiovascular development and diseases; 3. Park et al. 2018 (Roles of Hippo-YAP/TAZ in EC sprouting and junction maturation in angiogenesis) related to ours,  we believe our review is much more comprehensive and in-depth, and covers wider areas. Most significantly, we have the following novel topics and contents that are not covered by previous reviews:
         1) We described in details on the role of YAP/TAZ in many pathological angiogenesis such as diabetes, AMD, etc.
         2) We described the roles of YAP/TAZ and its upstream Hippo signaling in regulating various endothelial cell functions (e.g. endothelial-macrophage interactions and vascular differentiation) in angiogenesis.
         3) We provided new discoveries on the roles of YAP/TAZ and upstream Hippo signaling in tumor angiogenesis (e.g. Cho et al., 2019; Xu et al,, 2019; Zhu et al., 2018).
         4)  We provided recent discoveries on a new research area, i.e. the roles of YAP/TAZ in vascular mimicry.

Specific comments 

1. In general the text lacks references.  For example: 

* Line 110: …Since endothelial cells… this sentence should have references.

New references are added in our revised manuscript.

·       Paragraph starting in line 337: the authors cite only refs 79 and 80 for describing the expression pattern of yap and taz in endothelial cells. However, other studies have also shown the expression of these cofactors in endothelial cells and thus should be fitted as well. 

We went through all of the references and added more references when needed.

* Line 432: first sentence lacks references. 

Thanks for the comment. All references and text double-checked and about 30 new references were added when needed.  New added references were highlighted in blue.

2. The role of yap taz in pericytes and in lymphatic endothelial cells should also be described. 

The new section is added in the main text to cover these two areas.

3. Line 126: a part of the sentence seems to be missing.

Thanks for the comment. This sentence was double checked for its accuracy.

4. figure 2. polarity proteins and junction proteins should be defined and indicated with names. 

The detail names are added to the picture.

5. The section of angiogenesis in development is really small. As most of the function for yap and taz has been described during physiological developmental angiogenesis, this section should be expanded.  

This section is expanded in our revised manuscript.

6. The section of vascular mimicry should be expanded. 

The section of vascular mimicry is expanded.

Round 2

Reviewer 2 Report

Although the authors have address most of the previous comments, there are still minor things that need to be changed:

1, The authors write: "Additionally YAP/TAZ can also regulate endothelial cells and angiogenesis through regulating 0f none-endothelial cells such as pericytes also known as vascular smooth muscle cells or mural cells". This is not entirely correct as pericytes are a different cell type as smooth muscle cells. Please re-write.

2. The authors indicate that they have corrected or added references but for example - still they reference one study to indicate YAP/TAZ expression in endothelial cells (ref. 100). However, refs. 101 and 107 also show YAP/TAZ expression in endothelial cells as they should also be cited.

Author Response

Response to Reviewer#2 questions:

Comments and Suggestions for Authors

Although the authors have address most of the previous comments, there are still minor things that need to be changed: 

1, The authors write: "Additionally YAP/TAZ can also regulate endothelial cells and angiogenesis through regulating 0f none-endothelial cells such as pericytes also known as vascular smooth muscle cells or mural cells". This is not entirely correct as pericytes are a different cell type as smooth muscle cells. Please re-write. 

This sentence has been corrected into: “….such as pericytes as well as vascular smooth muscle cells or mural cells”.

2. The authors indicate that they have corrected or added references but for example - still they reference one study to indicate YAP/TAZ expression in endothelial cells (ref. 100). However, refs. 101 and 107 also show YAP/TAZ expression in endothelial cells as they should also be cited. 

References 101 and 107 have been added after Ref. 100 on Line 387.